# New Insights into the Phenotype Switching of Melanoma

**DOI:** 10.3390/cancers14246118

**Published:** 2022-12-12

**Authors:** Chiara Pagliuca, Luca Di Leo, Daniela De Zio

**Affiliations:** 1Melanoma Research Team, Danish Cancer Society Research Center, 2100 Copenhagen, Denmark; 2Department of Drug Design and Pharmacology, Faculty of Health and Medical Sciences, University of Copenhagen, 2200 Copenhagen, Denmark

**Keywords:** melanoma, phenotype switching, plasticity, resistance, dedifferentiation, non-mutational reprogramming, targeted therapies

## Abstract

**Simple Summary:**

Late-stage melanoma is one of the leading causes of death from skin cancers, as it frequently acquires resistance to standard therapies. Melanoma aggressiveness relies both on high intratumoral heterogeneity and on the capability of melanoma cells to switch among different differentiation phenotypes. Recently, melanoma plasticity has indeed been pinpointed as a main cause of resistance to standard therapies. Melanoma takes advantage of extrinsic reprogramming, a common feature exploited also by melanocytes, to promote tumor progression. Unfortunately, extrinsic factors and molecular mechanisms driving phenotype switching upon treatment are yet to be thoroughly characterized. The aim of this review is to support this field of research by providing brand-new insights into melanoma plasticity. Starting from the origin of phenotype switching, we will report up-to-date molecular players and extrinsic factors determining different transcriptional programs. Finally, the latest therapeutic strategies to tackle this mechanism of resistance will be discussed.

**Abstract:**

Melanoma is considered one of the deadliest skin cancers, partly because of acquired resistance to standard therapies. The most recognized driver of resistance relies on acquired melanoma cell plasticity, or the ability to dynamically switch among differentiation phenotypes. This confers the tumor noticeable advantages. During the last year, two new features have been included in the hallmarks of cancer, namely “Unlocking phenotypic plasticity” and “Non-mutational epigenetic reprogramming”. Such are inextricably intertwined as, most of the time, plasticity is not discernable at the genetic level, as it rather consists of epigenetic reprogramming heavily influenced by external factors. By analyzing current literature, this review provides reasoning about the origin of plasticity and clarifies whether such features already exist among tumors or are acquired by selection. Moreover, markers of plasticity, molecular effectors, and related tumor advantages in melanoma will be explored. Ultimately, as this new branch of tumor biology opened a wide landscape of therapeutic possibilities, in the final paragraph of this review, we will focus on newly characterized drugs targeting melanoma plasticity.

## 1. Introduction

Among skin tumors, melanoma is one of the rarest and deadliest, mostly because of high metastatic potential when not treated at early stages [1]. Melanoma originates from melanocytes, the cells highly specialized in UV protection, nested in the deepest layers of the epidermis proximal to the dermis and derived from neural crest cells [2].

Melanoma bears a high mutational burden. Genetic intra- and intertumor heterogeneity is one of the consequences of constant exposure of the epidermis to environmental stressors, including UV rays and untested compounds that, topically applied on the skin, are further absorbed by the epidermis [3,4,5,6]. Moreover, early mutations leading to cancer development frequently occur within the mitogen-activated protein kinase (MAPK) pathway [7]. A well-studied mutation is BRAF^V600E^ (v-raf murine sarcoma viral oncogene homolog B1), which is found in about 50% of melanomas and in ~80% of benign nevi [8,9]. Such mutation in the MAPK pathway is indeed fundamental for naevogenesis, giving a limited boost to the proliferation of melanocytes before the nevus enters a senescent-like status [7,10]. This quiescent status may be however overcome by telomerase reverse transcriptase (TERT) or cyclin-dependent kinase inhibitor 2A (CDKN2A) mutations [11].

However, not all melanomas bear mutations in the MAPK pathway. Nevertheless, this other subset is considered highly immunogenic, as a result of high mutational burden and consequent generation of neoantigens. Therefore, tumors that are not characterized by mutations in BRAF are frequently treated by immune checkpoint inhibitors (ICIs) or other types of immunotherapy [12]. Since 2011, a breakthrough concerning the approval of new treatments for patients with advanced-stage melanoma has occurred; in particular, both therapies targeting MAPK and immunotherapies have been largely investigated [13], but with limited efficacy in certain instances. Standard therapies already approved for the treatment of melanoma are reported in Table 1.

Beyond significant intratumor heterogeneity (ITH), mostly conferred by elevated mutational burden, phenotypic plasticity seems to play a pivotal role in determining melanoma progression and therapy resistance [14]. Recently, four new cancer-acquired capabilities, including “unlocking phenotype plasticity” and “non-mutational epigenetic reprogramming” [15], were added to the cancer hallmarks. Both hallmarks are strictly related since phenotype plasticity highly relies on entangled epigenetic changes and tumor microenvironment (TME) components, which are able to stabilize transcriptional programs [14].

The high plasticity of melanoma cells allows effortless phenotypic switching, which can converge on relapse to MAPK inhibitors or insensitivity to proinflammatory signaling resulting from immunotherapy [16]. As a consequence, treatment for advanced melanoma remains an unmet therapeutic need that must be addressed. Since resistance to standard therapies frequently occurs with time, focusing on specific effectors that mediate relapse may constitute a game changer.

The aim of this review is to summarize the latest and most relevant discoveries in the field of melanoma phenotype switching and to highlight new plausible targets to prevent plasticity-driven relapse to well-known therapies.

**Table 1 cancers-14-06118-t001:** List of clinical trials.

Target	Clinical trials	Objective or Overall Response Rate (%)	Median Progression-Free Survival (Months)	Most Common (Any Grade) Adverse Events
BRAF **	Dacarbazine vs. Vemurafenib (BRIM3 trial) [17,18]	5–9 vs. 48–57	1.6–1.7 vs. 5.3–6.9	Arthralgia, rash, fatigue, cuSCC *, keratoacanthoma, nausea, alopecia, diarrhea, neutropenia, abnormal liver function tests, and photosensitivity
	Dacarbazine vs. Dabrafenib (BREAK-3 trial) [19]	6 vs. 50	2.7 vs. 5.1	Skin-related toxic effects, fever, arthralgia, fatigue, and headache
BRAF + MEK	Vemurafenib vs. Vemurafenib plus Cobimetinib (coBRIM trial) [20]	50 vs. 70	7.2 vs. 12.3	Pyrexia and dehydration
	Dabrafenib or Vemurafenib vs. Dabrafenib plus Trametinib (COMBI-d and COMBI-v trials) [21,22]	51–53 vs. 64–69	7.3–8.8 vs. 11–11.4	Pyrexia, hyperkeratosis, cuSCC *, and keratoacanthoma
	Vemurafenib vs. Encorafenib plus Binimetinib (COLUMBUS trial) [23]	40.8 vs. 63.5	7.3 vs. 14.9	Nausea, diarrhea. and vomiting
Immune checkpoints	Dacarbazine vs. Nivolumab (CheckMate 066 trial) [24]	13.9 vs. 40	2.2 vs. 5.1	Fatigue, pruritus, and nausea
	Ipilimumab vs. Nivolumab vs. Nivolumab plus Ipilimumab (CheckMate 067 trial) [25]	19 vs. 44 vs. 58	2.9 vs. 6.9 vs. 11.5	Skin-related and gastrointestinal events
	Ipilimumab vs. Pembrolizumab every 2–3 weeks (KEYNOTE-006 trial) [26]	13 vs. 37–36	2.8 vs. 5.6–4.1	Fatigue, pruritus, diarrhea, and rash
	Dacarbazine vs. Dacarbazine plus Ipilimumab (CA184-024 trial) [27]	10.3 vs. 15.2	3 vs. 3	Elevation of alanine aminotransferase and aspartate aminotransferase levels, diarrhea, pruritus, and rash
BRAF + MEK + Immune Checkpoint	Vemurafenib plus Cobimetinib vs Atezolizumab plus Vemurafenib plus Cobimetinib (IMspire150) [28]	65 vs. 66.3	10.6 vs. 15.1	Blood creatinine phosphokinase increased, diarrhea, rash, arthralgia, pyrexia, alanine aminotransferase increased, and lipase increased

* cuSCC: cutaneous squamous cell carcinoma; ** BRAFi can cause development of cuSCC or keratoacanthomas due to the paradoxical activation of the MAPK pathway in cells harboring wild-type BRAF [29].

## 2. Phenotype Switching

In recent years, the concept of differentiation has evolved. The commitment of the cell is no longer considered a unidirectional process. It is rather seen as a highly reprogrammable state that can be easily influenced by environmental signaling. This ability to transform is nowadays considered a new hallmark of cancer, defined as cell plasticity, and is observed in a variety of tumors [15]. The phenomenon of plasticity can be determined both by cell-intrinsic properties, such as a mutation that contributes to an oncogenic phenotype, and cell-extrinsic features, which are determined by the influence of the surrounding microenvironment [30].

Since embryonic development, neural crest cells are highly motile cells that undergo differentiation only once they reach their specific location within the body and own migration and epithelial–mesenchymal transition (EMT) capabilities. These cells originate highly specialized and nonspecialized cells of the peripheral nervous system, such as Schwann cells, peripheral neurons, osteocytes, chondrocytes, adipocytes, smooth muscle cells, melanocytes, and keratinocytes. In 2021, Vidács and coworkers observed that human adult epidermal melanocytes grown in a medium lacking cholera toxin and tumor promoter 12-O-tetradecanoyl phorbol-13-acetate (PMA), became bipolar and unpigmented [31]. Molecularly, they revealed a spontaneous dedifferentiation process supported by a drop in the expression of differentiation markers, such as tyrosinase-related protein-1 (TRP-1) and mast/stem cell growth factor receptor Kit (c-Kit), along with sustained expression of genes involved in dedifferentiation, such as epidermal growth factor receptor (EGFR) and nestin [31].

The plasticity of melanoma can be classified at transcriptional levels; in the paragraph below, different transcriptional programs are reported based on specific markers of differentiation.

### 2.1. Transcriptional Programs of Differentiation

The state of differentiation of melanoma is usually categorized into 2 major transcriptional programs: the proliferative and the invasive states (Figure 1) [32]. These two programs have been highly characterized, and they are frequently driven by master regulators, favoring the generation of divergent transcriptional profiles. Well-established markers of “proliferative” phenotype are microphthalmia-associated transcription factor (MITF^High^) and AXL^Low^, the latter being considered one of the determinant markers for the dedifferentiation state. The proliferative state is recognized as a differentiated and epithelial-like phenotype. MITF is the master regulator of lineage commitment and pigmentation in melanocytes and induces the transcription of prodifferentiation genes such as premelanosome protein (*PMEL*), dopachrome tautomerase (*DCT*), tyrosinase (*TYR*), and melan-A (*MLANA*). Upstream activators of MITF are SRY-box transcription factor 10 (SOX10), paired box 3 (PAX3), CAMP-responsive element-binding protein (CREB), and endothelin receptor type B (EDNRB). They are frequently upregulated within the proliferative phenotype [33,34].

Recently, the role of spalt-like transcription factor 4 (SALL4), an epigenetic player that maintains melanocyte differentiation, has been explored in a Tyr::Nras^Q61K^; Cdkn2a^−/−^ melanoma mouse model [35]. This epigenetic factor is highly involved in primary melanoma formation and cooperates with histone deacetylase 2 (HDAC2) to favor a more proliferative state by epigenetically repressing genes of the invasive signature [35]. Similar to SALL4, YY1 is another transcription factor (TF) that promotes melanoma initiation and tumor growth while being averse to cancer invasiveness; indeed, mice conditionally depleted for Yy1 show an increase in metastasis formation [36].

On the other hand, the invasive phenotype is characterized by MITF^Low^/SOX10^Low^/AXL^High^. Epigenetic factors of the invasive transition are zinc finger E-box binding homeobox 1 (ZEB1), homeobox A1 (HOXA1), and BMI1. Among the cell-intrinsic properties supporting the invasive signature, BRN2 (brain-2) is a TF usually upregulated in dedifferentiated cells [37]. The expression of BRN2 and MITF is mutually exclusive. Indeed, BRN2 directly inactivates MITF transcription by binding to its promoter in 501-mel melanoma cells ectopically expressing Brn-2, whereas high levels of MITF allow the transcription of miR-211, which impairs BRN2 translation [38,39]. BRN2 can also activate the NOTCH signaling, leading to dedifferentiation via the downstream effector EZH2 (enhancer of zeste 2 polycomb repressive complex 2 subunit) as demonstrated through gene set enrichment analysis (GSEA) using The Cancer Genome Atlas (TCGA) dataset [40].

Of note, the oncogenic Yes1-associated transcriptional regulator (YAP)/tafazzin (TAZ) pathway also plays its role in the dedifferentiation process by directly regulating AXL transcriptionally [41]. When YAP/TAZ is free to enter the nucleus, it binds the TEA domain transcription factors (TEADs) and forms further complexes with the activator protein 1 (AP-1). The cooperation of these factors allows the transcription of genes involved in S-phase entry and mitosis [42]. More importantly for this context, however, some of the most relevant AP-1 transcription factors were measured both at the cell population and single-cell level in different melanoma lines using multiplexed transcriptional and protein measurements. Via multivariate statistical modeling, researchers were able to determine that among the AP-1 family members, some factors that sustain less differentiated programs exist, such as Fos-related antigen 1 (FRA1), phospho-FRA1^S265^, FRA2, and cJUN and phospho-cJUN^S73^ [43].

Inhibitor of differentiation protein 3 (ID3) is another player highly involved in phenotype switching; the transcription profiling of 21 BRAF-mutated melanomas revealed an upregulation of ID3 in BRAF inhibitor (BRAFi)-resistant tumors compared to treatment-naïves. ID3 upregulation has been confirmed in additional BRAFi-resistant lines, also correlating with the transcriptional repression of SOX10 and MITF and transition towards a more dedifferentiated state [44]. Moreover, a TF recently characterized as a regulator of phenotype switching is the Aryl hydrocarbon Receptor (AhR). Indeed, the interaction of AhR with SRC, as shown by proximity ligation assay, resulted in the phosphoactivation of SRC and FAK, eventually converging in the translocation of the YAP/TAZ complex to the nucleus [45,46], the role of which in these processes has already been discussed above.

Finally, recent findings suggest that a remarkable increase of fibroblast growth factor 7 (FGF7) in dedifferentiated nonmalignant melanocytes is a determinant factor involved in transferring melanosomes towards keratinocytes. However, additional studies are needed to fully understand the role of FGF7 in dedifferentiation [31].

### 2.2. New Progressive Model of Plasticity

Despite the previously defined features allowing the separation of the two well-established transcriptional states, phenotype switching is a fluctuant mechanism that can also rely on intermediate states of differentiation (Figure 1). Variable MITF expression levels can be a valid example of progressive dedifferentiation and can determine various outputs in terms of phenotypic state. The variability in MITF expression relies on the regulation of its transcription factors, post-translational modifications (PTMs) of the protein, and its competition with other TFs for common regulatory elements. MITF has been described as a rheostat model since its high activity promotes differentiation, whereas normal activity leads cells to proliferation. Low levels of MITF confer invasiveness and stemness properties, and its absence promotes senescence [33,47].

Since a dual classification of transcriptional states nowadays falls restrictive, new progressive differentiation subtypes have been described and subsequently listed: hyperdifferentiated cells, melanocytic proliferating cells, transitory-intermediate migrating cells, therapy-induced starved-like melanoma cells, neural crest stem cell-like (NCSC-like) cells, and MITF-negative undifferentiated cells, respectively (Figure 1) [16,48].

Melanocytic proliferating cells and MITF-undifferentiated cells reflect, respectively, proliferative and invasive transcriptional programs.

The hyperdifferentiated state is determined by short exposure to BRAFi. In this context, melanoma cell lines display strong melanocytic features, such as upregulation of melanocytic antigens MART-1 (also known as MLANA) and gp100 (also known as PMEL) and increase in MITF expression [16].

The transitory state has been identified as a subset of cells enriched for gene ontology (GO) terms associated with both differentiation and the neural crest phenotype. When expression patterns of some TFs and receptor tyrosine kinases (RTK) genes were evaluated, this state displayed MITF^High^/AXL^Low^/SOX10^High^ as the proliferative state and NGFR^High^ (neural growth factor receptor) levels as the NCSC-like state [48]. The transitory phenotype is moderately sensitive to targeted therapies and is stabilized by the underlying expression of TFs such as SOX6, nuclear factor of activated T cells 2 (NFATC2), early growth response 3 (EGR3), E74 like ETS transcription factor 1 (ELF1), and ETS variant transcription factor 4 (ETV4), as identified by inferring gene regulatory network on single-cell and bulk RNAseq data [49].

The transition towards starved-like melanoma cells occurs as a consequence of nutrient deprivation. This phenotype is characterized by peculiar metabolic alteration and a more slow-cycling dedifferentiated state [50]. Starved-like melanoma cells are usually no longer able to synthesize monounsaturated fatty acids (MUFA), thus requiring fatty acids uptaken from the TME, sustained by increased expression of CD36, a fatty acid transporter that determines a slow-cycling dedifferentiated phenotype [50,51]. This slow-cycling persistent state enables melanoma cells to survive selective pressure, and it can be easily switched off once environmental insults end, as it relies on epigenetic programs. Slow-cycling cells display high expression of H3K4 demethylase KDM5B (lysine demethylase 5B). KDM5B has been depicted as a protein harboring a “Janus-faced” role since it apparently impairs invasiveness and melanoma growth, but on the other side, it plays a fundamental role in allowing survival of melanoma to therapeutic insults, as shown in models induced to have a stable expression of KDM5B [52].

Longer exposure to BRAFi tends to stabilize the NCSC-like state [53]. Such phenotype is described as MITF^Low^ and is also known for its high expression of NGFR, SOX10, AXL, aquaporin 1 (AQP1), GDNF family receptor alpha 2 (GFRA2), and retinoid X receptor gamma (RXRγ) [48]. Besides well-established markers of NCSC-like status, a significant increase in expression of lysine demethylase 4B (Kdm4b) has been recently characterized in the Cancer Cell Line Encyclopedia (CCLE) dataset and other 21 human melanoma lines [54]. The NCSC-like phenotype also tends to acquire expression of neural markers, such as SOX2, SOX5, and, SOX8 [55].

Finally, the MITF-negative undifferentiated cells are prone to lose the expression of SOX10, hence acquiring a preneural crest phenotype upregulating SOX9, as demonstrated in MITF-methylated melanoma cells knocked out for SOX10 [55].

### 2.3. Phenotype Switching: An Acquired or Pre-Existing Feature?

Despite the scientific dispute regarding the existence of cancer stem cells (CSCs), scientific evidence has been highlighting that tumor growth relies on CSCs, at least in some cancers, such as breast cancer, colorectal cancer, brain cancer, and leukemia [56]. The hypothesis of a hierarchical organization started by CSCs in melanoma is still debated. In the meanwhile, several studies on patients with melanoma at different stages have shown that melanoma CSCs are the result of peculiar dynamic and reversible mechanisms, allowing phenotype transitions among differentiation states [57].

Over the years, the scientific community has been trying to determine the real origin of such plasticity, particularly whether it is induced by stressors and therapies, or if different phenotypes already exist in treatment-naïve tumors. In 2017, Ennen et al. found that treatment-naïve primary human melanoma already shows a predominant presence of MITF^High^ cells, sprinkled with a sporadic dedifferentiated population of AXL^High^/MITF^Low^ cells [58]. Such evidence was confirmed in 2018 by Rambow and coworkers, who observed the presence of different subpopulations (in terms of cell phenotypes) among drug-naïve lesions in patients [59]. Notably, a coexistence of states defined as proliferative, invasive, and NCSC was depicted [59]. Upon treatment with MAPK inhibitors (MAPKi), melanoma exacerbates some of these subclones, such as NCSCs, and further emerging clones defined as starved melanoma cells (SMCs), and MITF^High^ pigmented populations arise [59].

In 2017, Shaffer and coworkers, after monitoring cells exposed to MAPKi by long-term time-lapse imaging, hypothesized that phenotypic plasticity-driven resistance occurs in two stages: a rare subpopulation of cells expressing some resistant markers is already pre-resistant within the tumor and, upon drug exposure, these cells stabilize their resistant phenotype via transcriptome changes [60].

Thus, different phenotype states may coexist in melanoma; however, when exposed to environmental stressors or therapy insults, the tumors may employ their transcriptome complexity to favor the development or selection of drug-tolerant phenotypes. Despite the high genetic ITH of melanoma, most of the time resistance to therapies does not rely on genetic mutations. Two major mechanisms of resistance have been described: Darwinian selection, based on pre-existing subclones already resistant to treatments (given their genetic or epigenetic landscape), or Lamarckian induction, consisting of therapy-induced phenotypes that were not resistant to treatment a priori [61].

During Lamarckian induction, malignant cells exploit the ability to adapt to environmental cues without turning to genetic lesions in a dynamic and plastic way [61].

Although research has been conducted to identify potential effectors of plasticity, the major players have yet to be identified. However, phenotype transition can be stabilized with treatment, and numerous external factors that vary with drug exposure can converge to select a specific phenotype or facilitate phenotype switching.

For instance, proinflammatory response driven by immunotherapies leads to secretion of cytokines [16]. Tumor necrosis factor (TNF)-α has been identified as a key cytokine impacting melanocytic antigens and leading to the failure of adoptive T-cell transfer therapies (ACTs) in the transgenic mouse-derived HCmel3 melanoma cell line [62]. This cytokine induces downstream activation of the nuclear factor-κB (NFκB) and c-Jun pathways. Since c-Jun and MITF transcriptionally repress each other, TNF-α induces MITF loss favoring c-Jun induction and establishing a feed-forward mechanism nourishing MITF^Low^/c-Jun^High^ subclones. The latter phenotype allows myeloid cell infiltration counteracting immune responses towards immunotherapies [63].

Another key factor that varies upon immunotherapies is the environmental transforming growth factor-beta 1 (TGF-β). This can drive resistance by lowering expression of major histocompatibility complex class I (MHC-I) and limiting infiltration of T cells into the tumor. The presence of this soluble factor within the melanoma microenvironment also leads to the enrichment of the AXL^High^/MITF^Low^ phenotype [64].

A key role concerning therapy resistance is played by the cells composing the TME; indeed, MAPK inhibition also impacts the fibroblastic stroma. By using engineered cells expressing a kinase biosensor revealing MAPK pathway activity, researchers observed that BRAF inhibition leads to proliferation and paradoxical hyperactivation of extracellular signal-regulated kinase (ERK) signaling in BRAF wild-type cells [65,66]. Activation of the melanoma-associated fibroblasts (MAFs) leads to remodeling of the extracellular matrix (ECM) and increases its stiffness, converging on the integrin β1/FAK/Src signaling in melanoma cells and further translocation of YAP/TAZ to the nucleus. YAP signaling converges both on ERK activation, conferring melanoma a vertical resistance by downstream activation of the MAPK pathway, and on stabilization of a more dedifferentiated status [66,67].

Despite big scientific efforts to identify factors involved in treatment-induced dedifferentiation, the characterization of plasticity markers still results easier than determining specific molecules driving resistance. Extensive and in-depth analyses are still needed to determine who and what confers melanoma the ability to switch phenotypes.

## 3. Characteristics That Make Plasticity Advantageous for Melanoma

### 3.1. Reversibility

Melanoma plasticity mostly relies on non-mutational mechanisms driving phenotype switching, thus making it a reversible process. Melanoma cells sense the inputs from the TME and adapt their biological properties based on necessities. The advantage of non-mutational adaptations is that they are proportional to the stress/situation perceived, resulting in the activation of either short-term or stable transcriptional programs [59].

Treatment exposure leads to inevitable changes in chromatin conformation and stabilization of the transcriptional-resistant states. Assay for transposase-accessible chromatin using sequencing (ATAC-seq) analyses confirmed a high reshaping of TF-accessible sites [60]. Upon 4 weeks of treatment with vemurafenib (BRAFi), treated cells gain about 9000 TF accessible sites, determining a more open status of the chromatin. In particular, during the first week of treatment with MAPKi, an incredible loss of accessible sites for SOX10 binding and a gaining of accessible sites for TEAD and Jun/AP-1 occur [60].

Reversibility is a key feature employed by resistant cells to escape from treatments and metastasize to other tissues. Dedifferentiation has been reported as a transient response of adaptive resistance, providing the tumor a way out to acquire resistance [16].

The necessity of this dynamic switch between states has been proven by a study on NGFR^High^ cells. NGFR^High^ subpopulation allows the tumor to transiently escape from therapeutic insults. However, as demonstrated in xenograft mice injected with inducible NGFR-overexpressing cells, efficient metastasization to the lungs could not be achieved until NGFR was overexpressed. Thus, once reached the site of metastasis, the tumor recurs to its reversibility properties to switch towards a more differentiated phenotype and restore its proliferative features [68].

However, reversibility can also be a double-edged sword for the tumor. Indeed, despite the reversibility of epigenetic mutations, when a tumor is exposed to MAPK inhibitors for a prolonged time, a new phenotype is established, leading to adaptive resistance and to a phenomenon known as “drug addiction”. When such a phenotype is highly stable and the tumor becomes MAPKi-addicted, drug removal can result in cancer cell death since the cell is no longer able to switch quickly its stable phenotype to a different one, as demonstrated in melanoma cell lines, animal models, and patients [69].

### 3.2. Heterogeneity of Phenotypes: Treatment Failure and Metastasis

The dual classification of transcriptional cell states into melanocytic and invasive phenotypes is now considered restrictive and incomplete. Multiple transcriptional programs have been identified over time, as reported in the previous paragraphs, and different melanoma subpopulations can differently contribute to either tumor growth or metastatic dissemination.

Melanoma takes advantage of the heterogeneity of subpopulations with different tasks; for instance, there is a newly characterized population mainly devoted to promoting metastatic dissemination without playing any role in the expansion of the primary tumor [70]. These pre-migratory cells, defined as the pre-epithelial–mesenchymal transition (pre-EMT) population, were identified by performing clonal analysis on tumors grafted into syngeneic mouse models: such cells are supposed to function as “melanoma stem cells”. These pre-mesenchymal melanoma cells strictly associate with tumor vasculature and give rise to two different offspring: cells that maintain features similar to the mother and are able to further colonize the proximal vasculature, and another type of cell capable of detaching from the local niche and reaching distant sites. By predicting cell-to-cell communication within TME, scientists were able to characterize that stimulation and maintenance of the pre-EMT population are sustained by activation of NOTCH signaling mediated by delta-like 4 (DLL4) ligands present within endothelial cells (ECs) [70]. Thus, the cooperation of cells from TME is also pivotal for the maintenance of this transcriptional heterogeneity and cancer spread over the body.

Subpopulational heterogeneity is crucial for the establishment of more invasive phenotypes. Indeed, researchers assessed that MITF^High^ cells are fundamental for the outgrowth of the resistant population of slow-cycling AXL^High^ cells. Early upon BRAF inhibitor treatment, cells that show high levels of MITF start to secrete endothelin 1 (EDN1), a master regulator of heterogeneity. EDN1 plays a dual role since it is sensed both by AXL^High^ cells through the endothelin receptor type A (EDNRA) and by the MITF^High^ population through the EDNRB receptor. Downstream pathways activated by EDN1 lead to the expansion of BRAF-resistant AXL^High^ cells and to the protection of MITF^High^ cells from MAPKi. The latter was demonstrated in vivo since A375 cells knocked out for the EDNRB receptor and injected into the zebrafish xenograft model were not protected from BRAF inhibition [71].

Finally, heterogeneity conferred by plasticity seems to be involved in metastasis. Migration of clusters of cells has been observed in melanoma and results in efficient tumor dissemination and premetastatic conditioning. Heterogeneity among phenotypes in these clusters allows for the survival of migrating cells and easy seeding at the new site of metastasis, as demonstrated using circulating tumor cells derived from the blood sample of patients with metastatic cutaneous melanoma [72]. For instance, once detached from the primary site, cells must face anoikis, the stress of detachment; in this case, only cells harboring MITF^Low^ result resistant to detachment from the ECM [73]. Since MITF^High^ has been found in clusters of circulating tumors, Arozarena et al. hypothesized that MIFT^Low^ cells within metastatic clusters protect a subpopulation sensitive to anoikis [14,74].

### 3.3. Plasticity and TME Are Highly Interconnected

Based on their surroundings, melanoma cells sense external inputs and use dynamic switches among phenotypes to adapt to new environmental conditions; thus, plasticity is highly dependent on signals of the TME.

Different outcomes exploited by YAP signaling based on environmental components provide a good example of how easily TME influences phenotype switching. The YAP/TAZ pathway can be activated by different inputs, such as loss of cell polarity, mechanical forces, cell contact, and diffusible signals [75]. Activation of FAK1 is one of the leading causes contributing to the activation of the YAP/TAZ pathway and the following switching to less differentiated phenotypes [46,76].

Moreover, cancer-associated fibroblasts (CAFs) are able to suppress YAP/PAX activity by the secretion of TGF-β leading to the activation of the alternative YAP/TEAD/SMAD (sma- and mad-related protein) signaling pathway that promotes melanoma to dedifferentiation. In the absence of CAFs, YAP collaborates with PAX to express MITF, as supported by chromatin immunoprecipitation-PCR (ChIP-PCR) analysis showing an enrichment of YAP localization at the MITF promoter [67].

In addition, nutrient supply by the TME plays a central role in phenotype switching; indeed, glucose or glutamine starvation promotes the invasive phenotype [32,77]. Nutrient restriction, oxygen limitation, and an acidic TME lead to the upregulation of activating transcription factor 4 (ATF4), resulting in the transcriptional repression of MITF and an increase in the expression of AXL [32].

Indeed, MITF is considered a lineage-restricted regulator of stearoyl-CoA desaturase (SCD), sustaining cell growth and division through the production of necessary MUFAs [51]. Thus, through stable isotope labeling by amino acids in cell culture (SILAC) followed by liquid chromatography-mass spectrophotometry (LC-MS), researchers observed that low levels of MITF correspond to a drop in SCD and a great reshaping of cell metabolism. Consistent decrease of SCD activates a positive feedback loop driven by NFκB, establishing the MITF^Low^/AXL^High^ phenotype, typical of more dedifferentiated melanomas [51]. Once MUFA synthesis is decreased and it does not depend anymore on MITF regulation, metastatic outgrowth is sustained by fatty acid uptake from the microenvironment by standard or alternative fatty acid transporters such as CD36 or solute carrier family 27 member 1 (SLC27A1) [51].

However, the impact of TME on cancer plasticity is not unidirectional. Recently, a new mechanism capable of reshaping the response of the innate immune system in melanoma has been observed. In particular, natural killer (NK)-cell action is guided through recognition of a specific ligand expressed by melanocytes, MHC Class I polypeptide-related sequence A/B (MICA/B), mediated by its receptor natural-killer group 2 member D (NKG2D) [78]. MICA/B ligand can be cleaved by a sheddase known as ADAM metallopeptidase domain 10 (ADAM10) [78,79]. Since MITF directly regulates transcription of ADAM10, high levels of MITF within melanoma cells lead to an upregulation of this sheddase and a reduction of NK-cell recognition of the tumor [78].

## 4. Phenotype Switching as a Therapeutic Strategy Resource

As reported, cancer plasticity is a remarkable feature that confers cancer a great advantage. On the other hand, this mechanism offers enormous availability of new targets ready to be explored. Table 2 reports the degree of sensitivity displayed by different dedifferentiation states to standard MAPKi and novel drugs targeting phenotype switching.

By high-throughput profiling of human melanoma cell lines, researchers have assessed that every state of differentiation adapts to targeted therapy in a different fashion. For instance, a more undifferentiated phenotype is associated with incomplete inhibition of MAPKs, whereas an NCSC-like state adapts by reducing its MAPK signaling requirement. Thus, based on the dependence on the MAPK pathway, each state of differentiation can be tackled by different therapeutic compounds, and among them are promising epigenetic modulators [54]. In particular, BRAF-mutant melanoma cell lines expressing NGFR^Low^/AXL^High^ are sensitive to SP2509, a Kdm1a inhibitor, only if not pretreated with MAPKi, probably by using the MAPK pathway to restore cell senescence [54]. On the other hand, NCSC-like cells mutated for BRAF and expressing NGFR^High^/AXL^Low^ are more vulnerable to the combination of Kdm4b inhibitors with drugs targeting Braf and Mek [54]. Similar to Kdm4d inhibitors, bromodomain and extraterminal motif (BET) inhibitors, when combined with BRAF/MEK inhibitors, can also efficiently kill melanoma cell lines mutated for BRAF [54,80].

The use of these drugs seems very selective, with no effect on melanocytes that have not been transformed. However, epigenetic modulators are not limited to a specific target gene; epigenetic factors play a crucial role in the epigenetic homeostasis of the whole chromatin.

Another valid strategy to avoid melanoma plasticity involves targeting ECM components. Among all external factors affecting phenotype switching, the composition of the ECM can be determinant for the dedifferentiation process. Recently, a study identified that melanoma cells insensitive to BRAF inhibitors are characterized by a remarkable expression of the metalloproteinase MT1-MMP and other components of ECM, such as fibronectin and collagen. MT1-MMP is able to activate the survival signaling of melanoma by binding integrin β1 (ITGB1), enabling the downstream activation of the survival pathway [81]. One well-known downstream effector of ITGB1 is FAK1, which auto-phosphorylates at tyrosine 397, converging on pathways involved in dedifferentiation [81]. MT1-MMP inhibitor, ND322, once combined with BRAFi can re-sensitize resistant clones, as demonstrated in a patient-derived xenograft (PDX) model of BRAF-mutant K457-resistant cells [81].

Different strategies to directly target phenotypic switching are now emerging. Among these are downstream and upstream combined inhibition of BRN2 by small molecules targeting its effectors EZH2 or its transcriptional regulator NFATc2, respectively, as observed in NFAT2^+^ EZH2^+^ melanoma cell lines [40,82]. Moreover, enapotamab vedotin is an antibody conjugated with an antimitotic drug that selectively binds and kills AXL^High^ cells, as previously shown in PDXs [83]. Finally, many efforts have been made to target the NCSC-like population using RXRγ antagonists (HX531) [50], NGFR inhibitors (AG-879) [84], and FAK1 inhibitors (PF562271) [85]. Another important plausible strategy to target dedifferentiated cells relies on SRC inhibition via Dasatinib combined with BRAFi (Dabrafenib). According to recent findings, this last combination can double the overall survival rate of Mel006R BRAFi-resistant PDX when compared to BRAFi alone [45].

Moreover, a therapy to delay the onset of acquired resistance has been developed. Through the screening of pharmacological inhibitors, researchers have identified how to selectively target a resistant subpopulation of melanoma by combining birinapant, an inhibitor of cellular inhibitor of apoptosis 1/2 (cIAP1/2), with Braf/Mek inhibitors. Proteins of the cIAP family have a role in the inhibition of apoptosis, and they used to be overexpressed in SOX10-deficient cells. In A375 xenograft mouse models, the combination of birinapant plus MEK and BRAF inhibitors displayed a durable response with no recurrence after drug removal when compared to BRAFi plus MEKi alone [86].

Finally, targeting plasticity can enhance melanoma immunogenicity and immune checkpoint inhibitor function [87]. Recent publications reported SOX10 as a promising target, allowing the transformation of an immunological “cold” tumor into a “hot” one. Indeed, SOX10 regulates programmed death-ligand 1 (PD-L1) expression through the interferon regulatory factors IRF4-IRF1 axis. IRF1 determines tumor immunogenicity through induction of targets, such as antigen-presenting molecules (MHC class I, TAP, β2M) and ligands of immune inhibitory receptors (PD-L1 and PD-L2). SOX10 directly induces the transcription of IRF4, a negative regulator of IRF1; thus, its inhibition can enhance tumor responsiveness to anti-PD1 therapy, as observed in mice subcutaneously implanted with murine D4M melanoma cells [87].

New therapies to tackle phenotype switching are continuously being developed; however, the heterogeneity typically associated with plasticity makes melanoma a sneaky target. As a result, personalized medicine combined with these alternative therapies represents a potential option to challenge this tumor.

**Table 2 cancers-14-06118-t002:** Sensitivity of different states of differentiation to most common MAPK inhibitors and new compounds targeting phenotype switching.

State of Differentiation	Markersof Differentiation	Sensitivity to Current MAPK Therapies	New Drugs *	Effect of the Drugs
Hyperdifferentiated cells	MART-1^High^, gp100^High^, AXL^Low^ [88]	MAPKi tolerantDabrafenib (BRAFi) + Trametinib (MEKi) [16,50]	ACY-1215 (HDAC1i) + RGFP109 (HDAC3i) + anti-PD1 [87]	The combo of HDACs inhibitors antagonizes SOX10 expression to improve immunogenicity of “cold” tumors [87]
Melanocytic proliferating cells	MITF^High^, SOX10^High^, AXL^Low^ [48]	
Therapy sensitive [16]

Transitory-intermediate migrating cells	MITF^High^, NGFR^High^, SOX10^High^, AXL^Low^ [48]	
Low tolerance [16]

Therapy-induced starved-like melanoma cells	CD36^High^ [50], KDM5B^High^, MITF^Low^ [52]		TMECG (tyrosinase (TYR)-processed antimetabolic agent) + Cpd1 [52]	
Medium tolerancePLX4720 (BRAFi) + Cobinetinib (MEKi) [52], Dabrafenib (BRAFi) + Trametinib (MEKi) [16,50]	Cpd1 enhances KDM5B^High^ expression and sensitizes to TMECG [52]
NCSC-like cells	NGFR^High^, SOX10^High^, AXL^High^, RXRγ^High^, MITF^Low^ [48]	High tolerance due to reduced requirement of MAPK pathwayVemurafenib (BRAFi) alone or + Trametinib (MEKi) [54],Dabrafenib (BRAFi) + Trametinib (MEKi) [50,84,85]	Kdm4bi + BRAFi/MEKi [54]	Epigenetic modulator targeting Ngfr^High^ populations [54]
Enapotamab vedotin [83]	It kills selectively AXL^High^ cells [83]
RXRγ antagonists (HX531) + MAPKi [50]	It delays resistance onset and decreases accumulation of NCSC-like cells
NGFRi (AG-879) [84]	This drug has been reported to block NGFR [84]
FAK1 inhibitors (PF562271)+ RXRγ antagonists (HX531) + Dabrafenib/Trametinib [85]	Combination to target nongenetic resistance and efficiently remove NCSC-like cells [85]
MITF-negative undifferentiated cells	AXL^High^, SOX9^High^, MITF^Low^, SOX10^Low^ [3]	High tolerance due to incomplete inhibition of MAPK pathwayVemurafenib (BRAFi) alone or + Trametinib (MEKi) [54],Dabrafenib (BRAFi) + Trametinib (MEKi) [50,83,86]	Kdm1a inhibitor (SP2509) if not pretreated with MAPKi [54]	Epigenetic modulator targeting NGFR^Low^/AXL^High^ populations [54]
Enapotamab vedotin [83]	It kills selectively AXL^High^ cells [83]
Birinapant combined with BRAF/MEK inhibitors [86]	Birinapant is an inhibitor of cIAP1/2 able to kill SOX10-deficient cells [86]

* Enapotamab vedotin is in Phase II of clinical trials, whereas all the other new drugs targeting phenotype switching are in preclinical phase

## 5. Conclusions

Melanoma is one of the rarest and deadliest forms of skin cancer. If not treated at early stages, it tends to acquire a high metastatic potential and gains resistance to standard therapies [1]. Most of the time, the resistance is acquired through non-mutational mechanisms, resulting in the switching of cell phenotype (Figure 2). However, since reprogramming is a feature shared by both malignant and non-malignant melanocytes, resistant lines can already exist in treatment-naïve melanoma (Figure 2). As reported, when cells are exposed to treatments, they tend to establish a more resistant phenotype, while additional transient phenotypes emerge to challenge drug exposure. Reversibility, heterogeneity, and crosstalk with TME are hallmarks of plasticity that provide melanoma cells with significant metastatic potential and therapeutic resistance (Figure 2). Over the last decade, researchers have attempted to identify even more markers, as well as effectors of plasticity. Despite promising results shown by new compounds targeting the main players of plasticity, tumors remain challenging to treat due to acquired dynamic plasticity and heterogeneity. A promising solution relies on a combination of personalized medicine with novel drugs targeting molecular players of phenotype switching (Figure 2). In this review, we provided readers with brand-new insights into melanoma plasticity as well as ground-breaking treatments to tackle phenotype switching. We also emphasized the need to further develop this cancer biology field in order to identify new plausible therapeutic strategies.

## Figures and Tables

**Figure 1 cancers-14-06118-f001:**
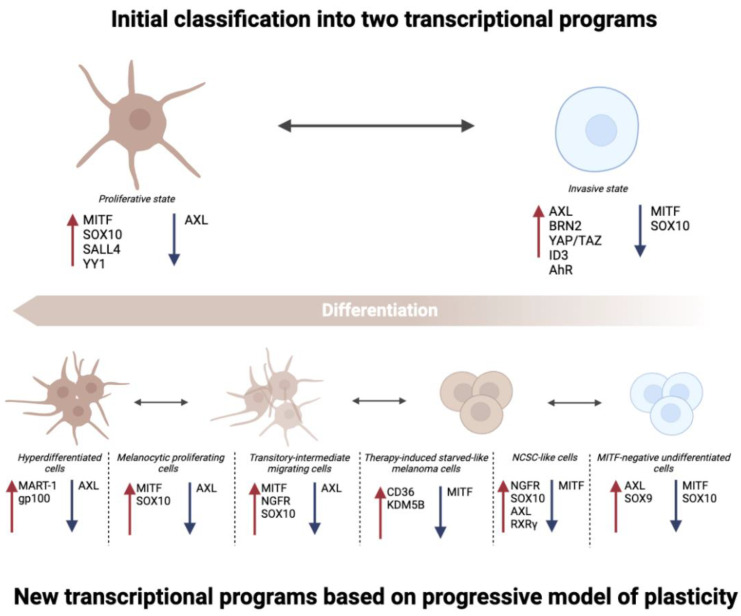
Different transcriptional programs have been defined throughout the years. Starting from an initial dual classification recognizing only a more differentiated “proliferative” state and a less differentiated “invasive” state, the range of transcriptional programs is nowadays increasing, following a progressive model of plasticity. So far, six transcriptional programs have been described: hyperdifferentiated cells, melanocytic proliferation cells, transitory-intermediate migrating cells, therapy-induced starved-like melanoma cells, NCSC-like cells, and MITF-negative undifferentiated cells. In the figure, all different transcriptional states are reported with their respective markers (created with BioRender.com).

**Figure 2 cancers-14-06118-f002:**
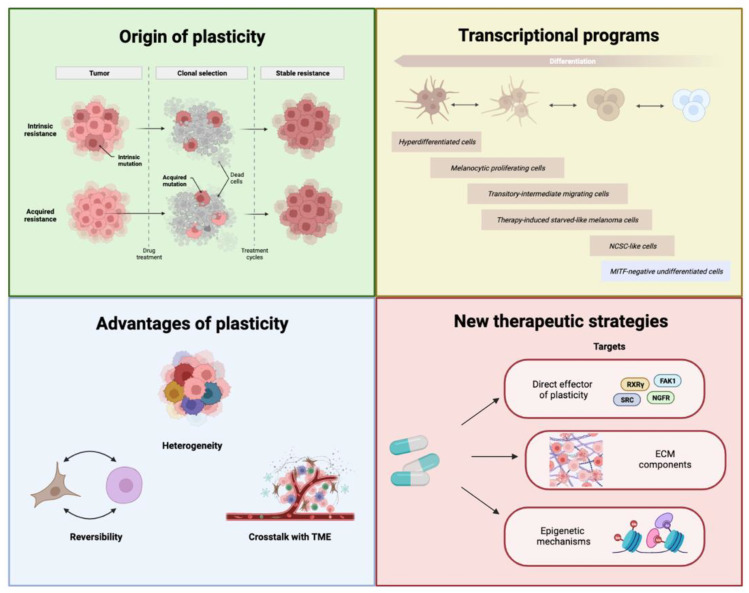
Summary of the different aspects of melanoma plasticity: origin of plasticity, transcriptional programs, advantages of plasticity, and new therapeutic strategies targeting plasticity (created with BioRender.com).

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
