# Peer review of "New Insights into the Phenotype Switching of Melanoma"

_cancers, 2022, doi:10.3390/cancers14246118_

Round 1

Reviewer 1 Report

The article is well written, well organized in its parts, and offers a comprehensive view of the plasticity and phenotypic switching in melanoma.

However, the title of the article suggests that the focus of the text will be represented by “therapeutic implications” that are completely absent in clinical practice. Moreover, the “main” argument cover just one paragraph in the article.

In addition, questionable informations regarding melanoma and its current prognosis are reported in several points.

For example:

Line 9: “minor percentage of patients survives therapy”: the percentage of “long survivor” patients treated with anti PD-1 and anti PD-1 plus anti CTLA-4 is about 40 and 50% respectively (data from the latest update of CheckMate 067).

Line 39: “melanoma represents the rarest”: melanoma is rarer than BCC and CSCC, but is not the rarest skin tumor neither the deadliest: Merkel Cell Carcinoma, porocarcinoma are rarer than melanoma and even more aggressive.

Line 72-74 “Since resistance to standard therapies inevitably occurs with time, focusing on specific effectors that mediate relapse may constitute a game changer.”: recurrence is actually not inevitable over time. Indeed, there are evidences that support the maintenance of the clinical response even after the interruption of the therapy (especially immunotherapy) in some subsets of patients.

 ..

In summary, I suggest the authors to change the title of the article, as the focus is not (it cannot be) the therapeutic implications of phenotypic switching, but the phenotypic switching and cellular plasticity in melanoma with a final chapter on future therapeutic possibilities.

I would also suggest to improve the clinical contents, explaining the efficacy and limitations of the therapies currently in use, before dedicating on the list of trials.

Finally, in this regard, I would suggest expanding the description of the experiments, better describing the obtained results.

Author Response

Reviewer #1:

The article is well written, well organized in its parts, and offers a comprehensive view of the plasticity and phenotypic switching in melanoma.

However, the title of the article suggests that the focus of the text will be represented by “therapeutic implications” that are completely absent in clinical practice. Moreover, the “main” argument cover just one paragraph in the article.

We are pleased that the Reviewer appreciated our manuscript and thank her/him for this suggestion. We have now changed the title with “New insights into the phenotype switching of melanoma”.

In addition, questionable informations regarding melanoma and its current prognosis are reported in several points.

For example:

Line 9: “minor percentage of patients survives therapy”: the percentage of “long survivor” patients treated with anti PD-1 and anti PD-1 plus anti CTLA-4 is about 40 and 50% respectively (data from the latest update of CheckMate 067).

We apologize to the Reviewer for the inaccuracy and in the new version of the manuscript we have rephrased it accordingly (line 10)

Line 39: “melanoma represents the rarest”: melanoma is rarer than BCC and CSCC, but is not the rarest skin tumor neither the deadliest: Merkel Cell Carcinoma, porocarcinoma are rarer than melanoma and even more aggressive.

We understand the Reviewer’s concern and do apologize for the inaccuracy. In the new version of the manuscript, we have rephrased it accordingly (lines 21, 47, 572).

Line 72-74 “Since resistance to standard therapies inevitably occurs with time, focusing on specific effectors that mediate relapse may constitute a game changer.”: recurrence is actually not inevitable over time. Indeed, there are evidences that support the maintenance of the clinical response even after the interruption of the therapy (especially immunotherapy) in some subsets of patients.

 ..

We understand the Reviewer’s concern and do apologize for the inaccuracy. Now we have formulated it differently, see line 83.

In summary, I suggest the authors to change the title of the article, as the focus is not (it cannot be) the therapeutic implications of phenotypic switching, but the phenotypic switching and cellular plasticity in melanoma with a final chapter on future therapeutic possibilities.

As mentioned above, we thank the Reviewer for this suggestion, and we have now changed the title with “New insights into the phenotype switching of melanoma”.

I would also suggest to improve the clinical contents, explaining the efficacy and limitations of the therapies currently in use, before dedicating on the list of trials.

We thank the Reviewer for giving us the opportunity to improve this aspect. We have now included a detailed table (Table 1) summarizing the standard melanoma therapies currently in use with additional details regarding their most common adverse events.

Finally, in this regard, I would suggest expanding the description of the experiments, better describing the obtained results.

We thank the Reviewer for this suggestion, and we have now revised the Manuscript accordingly (paragraphs 2, 3 and 4).

Reviewer 2 Report

This manuscript is a timely review on melanoma phenotype switching and its therapeutic implications. This review first summarizes phenotype switching highlighting the new transcriptional programs based on the progressive model of plasticity. Then the authors discuss whether phenotype switching is an acquired or a pre-existing feature. Moreover, they examine how melanoma cell plasticity fuels both metastasis and resistance to treatment and highlight the interconnection between plasticity and tumor microenvironment. Finally, the author discuss the therapeutic consequences of phenotype switching.

The subject is important, interesting and well presented by experts in the field. This topic is of interest to the readership of Cancers. Considering all the above achievements this review has reached, this article is suitable for publication with minor modifications. The manuscript could be improved by highlighting in a table or a figure the inhibitors which have been shown to target each state of differentiation and the sensitivity of these states to MAPK inhibitors. 

Author Response

Point-by-Point Rebuttal Letter

Reviewer #2:

This manuscript is a timely review on melanoma phenotype switching and its therapeutic implications. This review first summarizes phenotype switching highlighting the new transcriptional programs based on the progressive model of plasticity. Then the authors discuss whether phenotype switching is an acquired or a pre-existing feature. Moreover, they examine how melanoma cell plasticity fuels both metastasis and resistance to treatment and highlight the interconnection between plasticity and tumor microenvironment. Finally, the author discuss the therapeutic consequences of phenotype switching.

The subject is important, interesting and well presented by experts in the field. This topic is of interest to the readership of Cancers. Considering all the above achievements this review has reached, this article is suitable for publication with minor modifications. The manuscript could be improved by highlighting in a table or a figure the inhibitors which have been shown to target each state of differentiation and the sensitivity of these states to MAPK inhibitors. 

We are honoured that the Reviewer appreciated our manuscript and thank her/him for this suggestion to improve it. In the revised version of the manuscript, we have now included a detailed table (Table 2) that summarizes the sensitivity of the differentiation states to the most common MAPK inhibitors and the new compounds targeting phenotype switching in melanoma.

Round 2

Reviewer 1 Report

Thanks for understanding my suggestions.

I only recommend changing "standard therapies" to "clinical trials" in Table 1.